# Self-Paced Deep Reinforcement Learning

**Pascal Klink[1], Carlo D'Eramo[1], Jan Peters[1], Joni Pajarinen[1,2]**
[1] Intelligent Autonomous Systems, Technische Universität Darmstadt, Germany
[2] Department of Electrical Engineering and Automation, Aalto University, Finland
Correspondence to: `pascal.klink@tu-darmstadt.de`

## Abstract

Curriculum reinforcement learning (CRL) improves the learning speed and stability of an agent by exposing it to a tailored series of tasks throughout learning. Despite empirical successes, an open question in CRL is how to automatically generate a curriculum for a given reinforcement learning (RL) agent, avoiding manual design. In this paper, we propose an answer by interpreting the curriculum generation as an inference problem, where distributions over tasks are progressively learned to approach the target task. This approach leads to an automatic curriculum generation, whose *pace* is controlled by the agent, with solid theoretical motivation and easily integrated with deep RL algorithms. In the conducted experiments, the curricula generated with the proposed algorithm significantly improve learning performance across several environments and deep RL algorithms, matching or outperforming state-of-the-art existing CRL algorithms.

## 1 Introduction

Reinforcement learning (RL) [1] enables agents to learn sophisticated behaviors from interaction with an environment. Combinations of RL paradigms with powerful function approximators, commonly referred to as deep RL (DRL), have resulted in the acquisition of superhuman performance in various simulated domains [2, 3]. Despite these impressive results, DRL algorithms suffer from high sample complexity. Hence, a large body of research aims to reduce sample complexity by improving the explorative behavior of RL agents in a single task [4, 5, 6, 7].

Orthogonal to exploration methods, curriculum learning (CL) [8] for RL investigates the design of task sequences that maximally benefit the learning progress of an RL agent, by promoting the transfer of successful behavior between tasks in the sequence. To create a curriculum for a given problem, it is both necessary to define a set of tasks from which it can be generated and, based on that, specify *how* it is generated, i.e. how a task is selected given the current performance of the agent. This paper addresses the curriculum generation problem, assuming access to a set of parameterized tasks.

Recently, an increasing number of algorithms for curriculum generation have been proposed, empirically demonstrating that CL is an appropriate tool to improve the sample efficiency of DRL algorithms [9, 10]. However, these algorithms are based on heuristics and concepts that are, as of now, theoretically not well understood, preventing the establishment of rigorous improvements. In contrast, we propose to generate the curriculum based on a principled inference view on RL. Our approach generates the curriculum based on two quantities: The value function of the agent and the KL divergence to a target distribution of tasks. The resulting curriculum trades off task complexity (reflected in the value function) and the incorporation of desired tasks (reflected by the KL divergence). Our approach is conceptually similar to the self-paced learning (SPL) paradigm in supervised learning [11], which has only found application to RL in limited settings [12, 13].

**Contribution** We propose a new CRL algorithm, whose behavior is well explained as performing approximate inference on the common latent variable model (LVM) for RL [14, 15] (Section 4).

This enables principled improvements through the incorporation of advanced inference techniques. Combined with the well-known DRL algorithms TRPO, PPO and SAC [16, 17, 18], our method matches or surpasses the performance of state-of-the-art CRL methods in environments of different complexity and with sparse and dense rewards (Section 5).

## 2  Related Work

Simultaneously evolving the learning task with the learner has been investigated in a variety of fields ranging from behavioral psychology [19] to evolutionary robotics [20] and RL [21]. For supervised learning (SL), this principle was given the name *curriculum learning* [8]. This name has by now also been established in the RL community, where a variety of algorithms, aiming to generate curricula that maximally benefit the learner, have been proposed. Narvekar and Stone [22] showed that learning to create the *optimal* curriculum can be computationally harder than learning the entire task from scratch, motivating research on tractable approximations.

Keeping the agent's success rate within a certain range allowed to create curricula that drastically improve sample efficiency in tasks with binary reward functions or success indicators [23, 10, 24]. Many CRL methods [25, 26, 27, 28] have been proposed inspired by the idea of 'curiosity' or 'intrinsic motivation' [29, 30] – terms that refer to the way humans organize autonomous learning even in the absence of a task to be accomplished. Despite the empirical success, no theoretical foundation has been developed for the aforementioned methods, preventing principled improvements.

Another approach to curriculum generation has been explored under the name *self-paced learning* (SPL) for SL [11, 31, 32], proposing to generate a curriculum by optimizing the trade-off between exposing the learner to all available training samples and selecting samples in which it currently performs well. Despite its widespread application and empirical success in SL tasks, SPL has only been applied in a limited way to RL problems, restricting its use to the regression of the value function from an experience buffer [13] or to a strictly episodic RL setting [12]. Our method connects to this line of research, formulating the curriculum generation as a trade-off optimization of similar fashion. While the work by Ren et al. [13] is orthogonal to ours, we identify the result of Klink et al. [12] as a special case of our inference view. Besides allowing the combination of SPL and modern DRL algorithms to solve more complex tasks, the inference view presents a unified theory of using the self-paced learning paradigm for RL.

As we interpret RL from an inference perspective over the course of this paper, we wish to briefly point to several works employing this perspective [33, 34, 35, 15, 14]. Taking an inference perspective is beneficial when dealing with inverse problems or problems that require tractable approximations [36, 37]. For RL, it motivates regularization techniques such as the concept of maximum- or relative entropy [38, 18, 39] and stimulates the development of new, and interpretation of, existing algorithms from a common view [40, 41]. Due to a common language, algorithmic improvements on approximate inference [42, 43, 44] can be shared across domains.

## 3  Preliminaries

We formulate our approach in the domain of reinforcement learning (RL) for contextual Markov decision processes (CMDPs) [45, 46]. A CMDP is a tuple $<\mathcal{C}, \mathcal{S}, \mathcal{A}, \mathcal{M}>$, where $\mathcal{M}(\boldsymbol{c})$ is a function that maps a context $\boldsymbol{c} \in \mathcal{C}$ to a Markov decision process (MDP) $\mathcal{M}(\boldsymbol{c}) = <\mathcal{S}, \mathcal{A}, p_{\boldsymbol{c}}, r_{\boldsymbol{c}}, p_{0,\boldsymbol{c}}>$. An MDP is an abstract environment with states $\boldsymbol{s} \in \mathcal{S}$, actions $\boldsymbol{a} \in \mathcal{A}$, transition probabilities $p_{\boldsymbol{c}}(\boldsymbol{s}'|\boldsymbol{s}, \boldsymbol{a})$, reward function $r_{\boldsymbol{c}} : \mathcal{S} \times \mathcal{A} \mapsto \mathbb{R}$ and initial state distribution $p_{0,\boldsymbol{c}}(\boldsymbol{s})$. Typically $\mathcal{S}$, $\mathcal{A}$ and $\mathcal{C}$ are discrete spaces or subsets of $\mathbb{R}^n$. We can think of a CMDP as a parametric family of MDPs, which share the same state-action space. Such a parametric family of MDPs allows to share policies and representations between them [47], both being especially useful for CRL. RL for CMDPs encompasses approaches that aim to find a policy $\pi(\boldsymbol{a}|\boldsymbol{s}, \boldsymbol{c})$ which maximizes the expected return over trajectories $\tau = \{(\boldsymbol{s}_t, \boldsymbol{a}_t)|t \geq 0\}$

$$J(\mu, \pi) = E_{\mu(\boldsymbol{c}), p_{\pi}(\tau|\boldsymbol{c})} \left[ \sum_{t \geq 0} \gamma^t r_{\boldsymbol{c}}(\boldsymbol{s}_t, \boldsymbol{a}_t) \right], \quad p_{\pi}(\tau|\boldsymbol{c}) = p_{0,\boldsymbol{c}}(\boldsymbol{s}_0) \prod_{t \geq 0} p_{\boldsymbol{c}}(\boldsymbol{s}_{t+1}|\boldsymbol{s}_t, \boldsymbol{a}_t)\pi(\boldsymbol{a}_t|\boldsymbol{s}_t, \boldsymbol{c}), \quad (1)$$

with discount factor $\gamma \in [0, 1)$ and a probability distribution over contexts $\mu(\boldsymbol{c})$, encoding which contexts the agent is expected to encounter. We will often use the term policy and agent interchangeably,

as the policy represents the behavior of a (possibly virtual) agent. RL algorithms parametrize the policy $\pi$ with parameters $\boldsymbol{\omega} \in \mathbb{R}^n$. We will refer to this parametric policy as $\pi_{\boldsymbol{\omega}}$, sometimes replacing $\pi_{\boldsymbol{\omega}}$ by $\boldsymbol{\omega}$ in function arguments or subscripts, e.g. writing $J(\mu, \boldsymbol{\omega})$ or $p_{\boldsymbol{\omega}}(\tau|\boldsymbol{c})$. The so-called value function encodes the expected long-term reward when following a policy $\pi_{\boldsymbol{\omega}}$ starting in state $\boldsymbol{s}$

$$V_{\boldsymbol{\omega}}(\boldsymbol{s},\boldsymbol{c}) = E_{p_{\boldsymbol{\omega}}(\tau|\boldsymbol{s},\boldsymbol{c})}\left[\sum_{t\geq 0}\gamma^t r_{\boldsymbol{c}}(\boldsymbol{s}_t,\boldsymbol{a}_t)\right], \quad p_{\boldsymbol{\omega}}(\tau|\boldsymbol{s},\boldsymbol{c}) = \delta_{\boldsymbol{s}_0}^{\boldsymbol{s}}\prod_{t\geq 0}p_{\boldsymbol{c}}(\boldsymbol{s}_{t+1}|\boldsymbol{s}_t,\boldsymbol{a}_t)\pi_{\boldsymbol{\omega}}(\boldsymbol{a}_t|\boldsymbol{s}_t,\boldsymbol{c}), \quad (2)$$

where $\delta_{\boldsymbol{s}_0}^{\boldsymbol{s}}$ is the delta-distribution. The above value function for CMDPs has been introduced as a general or universal value function [48, 47]. We will, however, just refer to it as a value function, since a CMDP can be expressed as an MDP with extended state space. We see that the value function relates to Eq. 1 via $J(\mu, \boldsymbol{\omega}) = E_{\mu(\boldsymbol{c}),p_{0,\boldsymbol{c}}(\boldsymbol{s}_0)}\left[V_{\boldsymbol{\omega}}(\boldsymbol{s}_0,\boldsymbol{c})\right]$.

## 4 Self-Paced Deep Reinforcement Learning

Having established the necessary notation, we now introduce a curriculum to the contextual RL objective (Eq. 1) by allowing the agent to choose a distribution of tasks $p_{\boldsymbol{\nu}}(\boldsymbol{c})$, parameterized by $\boldsymbol{\nu} \in \mathbb{R}^m$, to train on. Put differently, we allow the RL agent to maximize $J(p_{\boldsymbol{\nu}}, \pi)$ under a chosen $p_{\boldsymbol{\nu}}$, only ultimately requiring it to match the "desired" task distribution $\mu(\boldsymbol{c})$ to ensure that the policy is indeed a local maximizer of $J(\mu, \pi)$. We achieve this by reformulating the RL objective as

$$\max_{\boldsymbol{\nu},\boldsymbol{\omega}} J(\boldsymbol{\nu}, \boldsymbol{\omega}) - \alpha D_{\mathrm{KL}}\left(p_{\boldsymbol{\nu}}(\boldsymbol{c})\|\mu(\boldsymbol{c})\right), \quad \alpha \geq 0. \quad (3)$$

In the above objective, $D_{\mathrm{KL}}\left(\cdot\|\cdot\right)$ is the KL divergence between two probability distributions. The parameter $\alpha$ controls the aforementioned trade-off between freely choosing $p_{\boldsymbol{\nu}}(\boldsymbol{c})$ and matching $\mu(\boldsymbol{c})$. When only optimizing Objective (3) w.r.t. $\boldsymbol{\omega}$ for a given $\boldsymbol{\nu}$, we simply optimize the contextual RL objective (Eq. 1) over the context distribution $p_{\boldsymbol{\nu}}(\boldsymbol{c})$. On the contrary, if Objective (3) is only optimized w.r.t. $\boldsymbol{\nu}$ for a given policy $\pi_{\boldsymbol{\omega}}$, then $\alpha$ controls the trade-off between incorporating tasks in which the policy obtains high reward and matching $\mu(\boldsymbol{c})$. So if we optimize Objective (3) in a block-coordinate ascent manner, we may use standard RL algorithms to train the policy under fixed $p_{\boldsymbol{\nu}}(\boldsymbol{c})$ and then adjust $p_{\boldsymbol{\nu}}(\boldsymbol{c})$ according to the obtained policy. If we keep increasing $\alpha$ during this procedure, $p_{\boldsymbol{\nu}}(\boldsymbol{c})$ will ultimately match $\mu(\boldsymbol{c})$ due to the KL divergence penalty and we train on the true objective. The benefit of such an interpolation between task distributions under which the agent initially performs well and $\mu(\boldsymbol{c})$ is that the agent may be able to adapt well-performing behavior as the environments gradually transform. This can, in turn, avoid learning poor behavior and increase learning speed. The outlined idea resembles the paradigm of self-paced learning for supervised learning [11], where a regression- or classification model as well as its training set are alternatingly optimized. The training set for a given model is chosen by trading-off favoring samples under which the model has low prediction error and incorporating all samples in the dataset. Indeed, the idea of generating curricula for RL using the self-paced learning paradigm has previously been investigated by Klink et al. [12]. However, they investigate the curriculum generation only in the episodic RL setting and jointly update the policy and context distribution. This ties the curriculum generation to a specific (episodic) RL algorithm, that, as we will see in the experiments, is not suited for high-dimensional policy parameterizations. Our formulation is not limited to such a specific setting, allowing to use the resulting algorithm for curriculum generation with any RL algorithm. Indeed, we will now relate the maximization of Objective (3) w.r.t. $\boldsymbol{\nu}$ to an inference perspective, showing that our formulation explains the results obtained by Klink et al. [12].

**Interpretation as Inference** Objective (3) can be motivated by taking an inference perspective on RL [14]. In this inference perspective, we introduce an 'optimality' event $\mathcal{O}$, whose probability of occurring is defined via a monotonic transformation $f : \mathbb{R} \mapsto \mathbb{R}_{\geq 0}$ of the cumulative reward $R(\tau, \boldsymbol{c}) = \sum_{t\geq 0} r_{\boldsymbol{c}}(\boldsymbol{s}_t, \boldsymbol{a}_t)$, yielding the following latent variable model (LVM)

$$p_{\boldsymbol{\nu},\boldsymbol{\omega}}(\mathcal{O}) = \int p_{\boldsymbol{\nu},\boldsymbol{\omega}}(\mathcal{O}, \tau, \boldsymbol{c})d\tau d\boldsymbol{c} \propto \int f(R(\tau,\boldsymbol{c}))p_{\boldsymbol{\omega}}(\tau|\boldsymbol{c})p_{\boldsymbol{\nu}}(\boldsymbol{c})d\tau d\boldsymbol{c}. \quad (4)$$

Under appropriate choice of $f(\cdot)$ and minor modification of the transition dynamics to account for the discounting factor $\gamma$ [15, 14], maximizing LVM (4) w.r.t. $\boldsymbol{\omega}$ is equal to the maximization of $J(p_{\boldsymbol{\nu}}, \pi_{\boldsymbol{\omega}})$ w.r.t. $\pi_{\boldsymbol{\omega}}$. This setting is well explored and allowed to identify various RL algorithms as

approximate applications of the expectation maximization (EM) algorithm to LVM (4) to maximize $p_{\nu,\omega}(\mathcal{O})$ w.r.t $\omega$ [40]. Our idea of maximizing Objective (3) in a block-coordinate ascent manner is readily supported by the EM algorithm, since its steps can be executed alternatingly w.r.t. $\nu$ and $\omega$. Consequently, we now investigate the case when maximizing $p_{\nu,\omega}(\mathcal{O})$ w.r.t. $\nu$, showing that known regularization techniques for approximate inference motivate Objective (3). For brevity, we only state the main results here and refer to Appendix A for detailed proofs and explanations of EM.

When maximizing $p_{\nu,\omega}(\mathcal{O})$ w.r.t. $\nu$ using EM, we introduce a variational distribution $q(c)$ and alternate between the so called E-Step, in which $q(c)$ is found by minimizing $D_{\mathrm{KL}}(q(c)\|p_{\nu,\omega}(c|\mathcal{O}))$, and the M-Step, in which $\nu$ is found by minimizing the KL divergence to the previously obtained variational distribution $D_{\mathrm{KL}}(q(c)\|p_\nu(c))$. Typically $q(c)$ is not restricted to a parametric form and hence matches $p_{\nu,\omega}(c|\mathcal{O})$ after the E-Step. We now state our main theoretical insight, showing exactly what approximations and modifications to the regular EM algorithm are required to retrieve Objective 3.

**Theorem 1** *Choosing* $f(\cdot)=\exp(\cdot)$, *maximizing Objective (3) minus a KL divergence term* $D_{KL}(p_\nu(c)\|p_{\tilde{\nu}}(c))$ *is equal to executing E- and M-Step while restricting $q(c)$ to be of the same parametric form as $p_\nu(c)$ and introducing a regularized E-Step* $D_{KL}\left(q(c)\left\|\frac{1}{Z}p_{\tilde{\nu},\omega}(c|\mathcal{O})^{\frac{1}{1+\alpha}}\mu(c)^{\frac{\alpha}{1+\alpha}}\right.\right)$.

Theorem 1 is interesting for many reasons. Firstly, the extra term in Objective (3) can be identified as a regularization term, which penalizes a large deviation of $p_\nu(c)$ from $p_{\tilde{\nu}}(c)$. In the algorithm we propose, we replace this penalty term by a constraint on the KL divergence between successive context distributions, granting explicit control over their dissimilarity. This is beneficial when estimating expectations in Objective (3) by a finite amount of samples. Next, restricting the variational distribution to a parametric form is a known concept in RL. Abdolmaleki et al. [40] have shown that it yields an explanation for the well-known on-policy algorithms TRPO and PPO [16, 17]. Finally, the regularized E-Step fits $q(c)$ to a distribution that is referred to as a *tempered* posterior. Tempering, or *deterministic annealing*, is used in variational inference to improve the approximation of posterior distributions by gradually moving from the prior (in our case $\mu(c)$) to the true posterior (here $p_{\nu,\omega}(c|\mathcal{O})$) [44, 49, 50], which in above equation corresponds to gradually decreasing $\alpha$ to zero. We, however, increasingly enforce $p_\nu(c)$ to match $\mu(c)$ by gradually increasing $\alpha$. To understand this "inverse" behavior, we need to remember that the maximization w.r.t. $\nu$ solely aims to generate context distributions $p_\nu(c)$ that facilitate the maximization of $J(\mu,\pi_\omega)$ w.r.t. $\omega$. This means to initially encode contexts in which the event $\mathcal{O}$ is most likely, i.e. $p_\omega(\mathcal{O}|c)$ is highest, and only gradually match $\mu(c)$. To conclude this theoretical section, we note that the update rule proposed by Klink et al. [12] can be recovered from our formulation.

**Theorem 2** *Choosing* $f(\cdot)=\exp(\cdot/\eta)$, *the (unrestricted) variational distribution after the regularized E-Step is given by* $q(c) \propto p_\nu(c) \exp\left(\frac{V_\omega(c)+\eta\alpha(\log(\mu(c))-\log(p_\nu(c)))}{\eta+\eta\alpha}\right)$, *where $V_\omega(c)$ is the 'episodic value function' as defined in [34].*

The variational distribution in Theorem 2 resembles the results in [12] with the only difference that $\alpha$ is scaled by $\eta$. Hence, for a given schedule of $\alpha$, we simply need to scale every value in this schedule by $1/\eta$ to match the results from Klink et al. [12].

**Algorithmic Realization** As previously mentioned, we maximize Objective (3) in a block-coordinate ascent manner, i.e. use standard RL algorithms to optimize $J(p_{\nu_i},\pi_\omega)$ w.r.t. $\pi_\omega$ under the current context distribution $p_{\nu_i}$. Consequently, we only need to develop ways to optimize Objective (3) w.r.t. $p_\nu$ for a given policy $\pi_{\omega_i}$. We can run any RL algorithm to generate a set of trajectories $\mathcal{D}_i = \left\{(c^k,\tau^k)\big|k \in [1,K], c_k \sim p_{\nu_i}(c), \tau_k \sim \pi_{\omega_i}(\tau|c_k)\right\}$ alongside an improved policy $\pi_{\omega_{i+1}}$. Furthermore, most state-of-the-art RL algorithms fit a value function $V_{\omega_{i+1}}(s,c)$ while generating the policy $\pi_{\omega_{i+1}}$. Even if the employed RL algorithm does not generate an estimate of $V_{\omega_{i+1}}(s,c)$, it is easy to compute one using standard techniques. We can exploit the connection between value function and RL objective $J(p,\omega_{i+1}) = E_{p(c),p_{0,c}(s_0)}\left[V_{\omega_{i+1}}(s_0,c)\right]$ to optimize

$$\max_{\nu_{i+1}} \frac{1}{K}\sum_{k=1}^{K} \frac{p_{\nu_{i+1}}(c^k)}{p_{\nu_i}(c^k)}V_{\omega_{i+1}}\left(s_0^k,c^k\right) - \alpha_i D_{\mathrm{KL}}\left(p_{\nu_{i+1}}(c)\big\|\mu(c)\right) \quad \text{s.t. } D_{\mathrm{KL}}\left(p_{\nu_{i+1}}(c)\big\|p_{\nu_i}(c)\right) \leq \epsilon.$$

$$(5)$$

**Algorithm 1** Self-Paced Deep Reinforcement Learning

---

**Input:** Initial context distribution- and policy parameters $\boldsymbol{\nu}_0$ and $\boldsymbol{\omega}_0$, Target context distribution $\mu(\boldsymbol{c})$, KL penalty proportion $\zeta$, offset $N_\alpha$, number of iterations $N$, Rollouts per policy update $K$
**for** $i = 1$ **to** $N$ **do**
    **Agent Improvement:**
    Sample contexts: $\boldsymbol{c}^k \sim p_{\boldsymbol{\nu}_i}(\boldsymbol{c}), \ k = 1, \ldots, K$
    Rollout trajectories: $\tau^k \sim \pi_{\boldsymbol{\omega}_i}(\tau | \boldsymbol{c}_k), \ k = 1, \ldots, K$
    Obtain $\pi_{\boldsymbol{\omega}_{i+1}}$ from RL algorithm of choice using $\mathcal{D}_i = \left\{ (\boldsymbol{c}^k, \tau^k) | k = 1, \ldots, K \right\}$
    Estimate $V_{\boldsymbol{\omega}_{i+1}}(\boldsymbol{s}_0^k, \boldsymbol{c}^k)$ for contexts $\boldsymbol{c}^k$ (using the employed RL algorithm, if possible)
    **Context Distribution Update:**
    Obtain $p_{\boldsymbol{\nu}_{i+1}}$ optimizing (Eq. 5), using $\alpha_i = 0, \ \text{if} \ i \leq N_\alpha, \ \text{else} \ \mathcal{B}(\boldsymbol{\nu}_i, \mathcal{D}_i)$ (Eq. 6)
**end for**

---

instead of Objective (3) to obtain $\boldsymbol{\nu}_{i+1}$. The first term in Objective (5) is an importance-weighted approximation of $J(\boldsymbol{\nu}_{i+1}, \boldsymbol{\omega}_{i+1})$. Motivated by Theorem 1, the KL divergence constraint between subsequent context distributions $p_{\boldsymbol{\nu}_i}(\boldsymbol{c})$ and $p_{\boldsymbol{\nu}_{i+1}}(\boldsymbol{c})$ avoids large jumps in the context distribution. Above objective can be solved using any constrained optimization algorithm. In our implementation, we use the trust-region algorithm implemented in the SciPy library [51]. In each iteration, the parameter $\alpha_i$ is chosen such that the KL divergence penalty w.r.t. the current context distribution is in constant proportion $\zeta$ to the average reward obtained during the last iteration of policy optimization

$$\alpha_i = \mathcal{B}(\boldsymbol{\nu}_i, \mathcal{D}_i) = \zeta \frac{\frac{1}{K} \sum_{k=1}^K R(\tau^k, \boldsymbol{c}^k)}{D_{\text{KL}}(p_{\boldsymbol{\nu}_i}(\boldsymbol{c}) \| \mu(\boldsymbol{c}))}, \quad R(\tau^k, \boldsymbol{c}^k) = \sum_{t \geq 0} \gamma^t r_{\boldsymbol{c}^k}(\boldsymbol{s}_t^k, \boldsymbol{a}_t^k), \tag{6}$$

as proposed by Klink et al. [12]. We further adopt their strategy of setting $\alpha$ to zero for the first $N_\alpha$ iterations. This allows to taylor the context distribution to the learner in early iterations, if the initial context distribution is uninformative, i.e. covers large parts of the context space. Note that this is a naive choice, that nonetheless worked sufficiently well in our experiments. At this point, the connection to tempered inference allows for principled future improvements by using more advanced methods to choose $\alpha$ [44]. For the experiments, we restrict $p_{\boldsymbol{\nu}}(\boldsymbol{c})$ to be Gaussian. Consequently, Objective (5) is optimized w.r.t. the mean $\boldsymbol{\mu}$ and covariance $\boldsymbol{\Sigma}$ of the context distribution. Again, the inference view readily motivates future improvements by using advanced sampling techniques [43, 42]. These techniques allow to directly sample from the variational distribution $q(\boldsymbol{c})$ in Theorem 1, bypassing the need to fit a parametric distribution and allowing to represent multi-modal distributions. The outlined method is summarized in Algorithm 1.

## 5 Experiments

The aim of this section is to investigate the performance and versatility of the proposed curriculum reinforcement learning algorithm (SPDL). To accomplish this, we evaluate SPDL in three different environments with different DRL algorithms to test the proposition that the learning scheme benefits the performance of various RL algorithms. We evaluate the performance using TRPO [16], PPO [17] and SAC [18]. For all DRL algorithms, we use the implementations provided in the `Stable Baselines` library [52]. [1]

The first two environments aim at investigating the benefit of SPDL when the purpose of the generated curriculum is solely to facilitate the learning

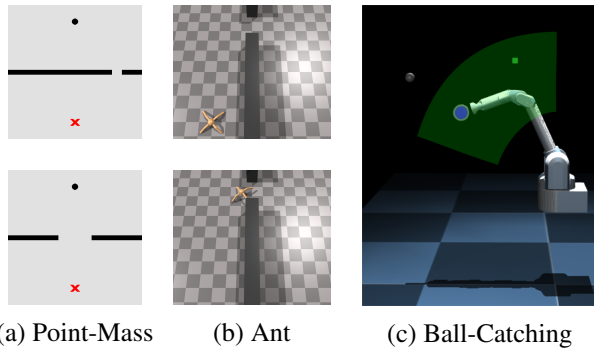

(a) Point-Mass     (b) Ant     (c) Ball-Catching

Figure 1: Environments used for experimental evaluation. For the point mass environment (a), the upper plot shows the target task. The shaded areas in picture (c) visualize the target distribution of ball positions (green) as well as the ball positions for which the initial policy succeeds (blue).

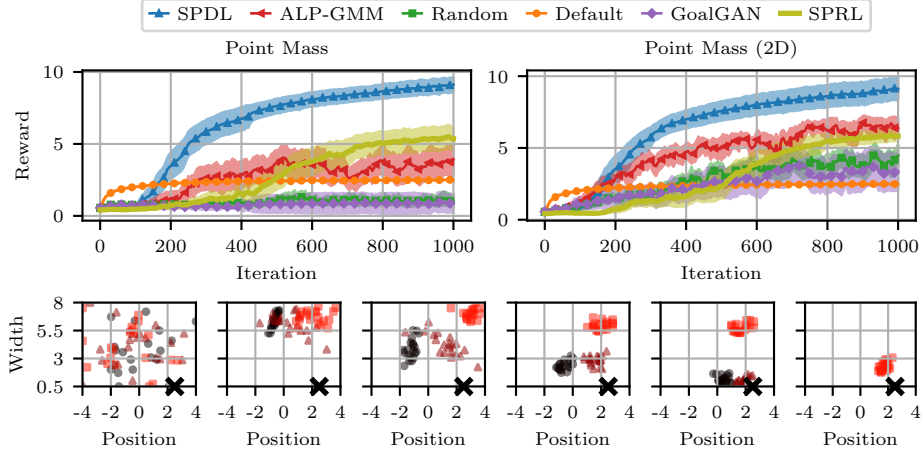

Figure 2: Reward of different curricula in the Point Mass (2D and 3D) environment for TRPO. Mean (thick line) and two times standard error (shaded area) is computed from 20 algorithm runs. The lower plots show samples from the context distributions $p(\boldsymbol{c})$ in the Point-Mass 2D environment at iterations 0, 50, 80, 100, 120 and 200 (from left to right). Different colors and shapes of samples indicate different algorithm runs. The black cross marks the mean of the target distribution $\mu(\boldsymbol{c})$.

of a hard target task, which the agent is not able to solve without a curriculum. For this purpose, we create two environments that are conceptually similar to the point mass experiment considered by SPRL [12]. The first one is a copy of the original experiment, but with an additional parameter to the context space, as we will detail in the corresponding section. The second environment extends the original experiment by replacing the point mass with a torque-controlled quadruped 'ant'. This increases the complexity of the underlying control problem, requiring the capacity of deep neural network function approximators used in DRL algorithms. The final environment is a robotic ball-catching environment. This environment constitutes a shift in curriculum paradigm as well as reward function. Instead of guiding learning towards a specific target task, this third environment requires to learn a ball-catching policy over a wide range of initial states (ball position and velocity). The reward function is sparse compared to the dense ones employed in the first two environments.

To judge the performance of SPDL, we compare the obtained results to state-of-the-art CRL algorithms ALP-GMM [27], which is based on the concept of Intrinsic Motivation, GoalGAN [23], which relies on the notion of a success indicator to define a curriculum, and SPRL [12], the episodic counterpart of our algorithm. Furthermore, we compare to curricula consisting of tasks uniformly sampled from the context space (referred to as 'Random' in the plots) and learning without a curriculum (referred to as 'Default'). Additional details on the experiments as well as qualitative evaluations of them can be found in Appendix B.

## 5.1 Point Mass Environment

In this environment, the agent controls a point mass that needs to be navigated through a gate of given size and position to reach a desired target in a two-dimensional world. If the point mass crashes into the wall, the experiment is stopped. The agent moves the point mass by applying forces and the reward decays in a squared exponential manner with increasing distance to the goal. In our version of the experiment, the contextual variable $\boldsymbol{c} \in \mathbb{R}^3$ changes the width and position of the gate as well as the dynamic friction coefficient of the ground on which the point mass slides. The target context distribution $\mu(\boldsymbol{c})$ is a narrow Gaussian with negligible noise that encodes a small gate at a specific position and a dynamic friction coefficient of 0. Figure 1 shows two different instances of the environment, one of them being the target task.

Figure 2 shows the results of two different experiments in this environment, one where the curriculum is generated over the full context space and one in which the friction parameter is fixed to its target value of 0. As Figure 2 and Table 1 indicate, SPDL significantly increases the asymptotic reward on the target task compared to all other methods. Furthermore, we see that SPRL, which we applied by defining the episodic RL policy $p(\boldsymbol{\omega}|\boldsymbol{c})$ to choose the weights $\boldsymbol{\omega}$ of the policy network for a

Table 1: Average final reward and standard error of different curricula and RL algorithms in the two Point Mass environments with three (P3D) and two (P2D) context dimensions as well as the Ball-Catching environment (BC). The data is computed from 20 algorithm runs. Significantly better results according to a t-test with $p < 1\%$ are highlighted in bold. The asterisks mark runs of SPDL/GoalGAN with an initialized context distribution and runs of Default learning without policy initialization.

|  | PPO (P3D) | SAC (P3D) | PPO (P2D) | SAC (P2D) | TRPO (BC) | PPO (BC) |
|---|---|---|---|---|---|---|
| ALP-GMM | $2.34 \pm 0.2$ | $0.96 \pm 0.3$ | $5.24 \pm 0.4$ | $1.15 \pm 0.4$ | $39.8 \pm 1.1$ | $46.5 \pm 0.7$ |
| GoalGAN | $0.50 \pm 0.0$ | $1.08 \pm 0.4$ | $1.39 \pm 0.5$ | $0.72 \pm 0.2$ | $42.5 \pm 1.6$ | $42.6 \pm 2.7$ |
| GoalGAN* | - | - | - | - | $45.8 \pm 1.0$ | $45.9 \pm 1.0$ |
| SPDL | $\mathbf{9.35 \pm 0.1}$ | $\mathbf{4.43 \pm 0.7}$ | $\mathbf{9.02 \pm 0.4}$ | $\mathbf{4.69 \pm 0.7}$ | $47.0 \pm 2.0$ | $\mathbf{53.9 \pm 0.4}$ |
| SPDL* | - | - | - | - | $43.3 \pm 2.0$ | $49.3 \pm 1.4$ |
| Random | $0.53 \pm 0.0$ | $0.60 \pm 0.1$ | $1.34 \pm 0.3$ | $0.93 \pm 0.3$ | - | - |
| Default | $2.46 \pm 0.0$ | $2.26 \pm 0.0$ | $2.47 \pm 0.0$ | $2.23 \pm 0.0$ | $21.0 \pm 0.3$ | $22.1 \pm 0.3$ |
| Default* | - | - | - | - | $21.2 \pm 0.3$ | $23.0 \pm 0.7$ |

given context $c$, also leads to a good performance. Increasing the dimension of the context space has a stronger negative impact on the performance of the other CL algorithms than on both SPDL and SPRL, where it only negligibly decreases the performance. We suspect that this effect arises because both ALP-GMM and GoalGAN have no notion of a target distribution. Consequently, for a context distribution $\mu(c)$ with negligible variance, a higher context dimension decreases the average proximity of sampled tasks to the target one. By having a notion of a target distribution, SPDL ultimately samples contexts that are close to the desired ones, regardless of the dimension. The context distributions $p(c)$ visualized in Figure 2 show that the agent focuses on wide gates in a variety of positions in early iterations. Subsequently, the size of the gate is decreased and the position of the gate is shifted to match the target one. This process is carried out at different pace and in different ways, sometimes preferring to first shrink the width of the gate before moving its position while sometimes doing both simultaneously.

## 5.2 Ant Environment

We replace the point mass in the previous environment with a four-legged ant similar to the one in the OpenAI Gym simulation environment [53]. [2] The goal is to reach the other side of a wall by passing through a gate, whose width and position is determined by the contextual variable $c \in \mathbb{R}^2$ (Figure 1).

Opposed to the previous environment, an application of SPRL is not straightforward in this environment, since the episodic policy needs to choose weights for a policy network with $6464$ parameters. In such high-dimensional spaces, fitting the new episodic policy (i.e. a $6464$-dimensional Gaussian) to the generated samples requires significantly more computation time than an update of a step-based policy, taking up to 25 minutes per update on our hardware. Furthermore, this step is prone to numerical instabilities due to the large covariance matrix that needs to be estimated. This observation stresses the benefit of our CRL approach, as it unifies the curriculum generation for episodic and step-based RL algorithms, allowing to choose the most beneficial one for the task at hand.

In this environment, we were only able to evaluate the CL algorithms using PPO. This is because the implementations of TRPO and SAC in the `Stable-Baselines` library do not allow to make use of the parallelization capabilities of the Isaac Gym simulator, leading to prohibitive running times (details in Appendix B).

Looking at Figure 3, we see that SPDL allows the learning agent to escape the local optimum which results from the agent not finding the gate to pass through. ALP-GMM and a random curriculum do not improve the reward over directly learning on the target task. However, as we show in Appendix B, both ALP-GMM and a random curriculum improve the qualitative performance, as they sometimes allow to move the ant through the gate. Nonetheless, this behavior is unreliable and inefficient, causing the action penalties in combination with the discount factor to prevent this better behavior from being reflected in the reward.

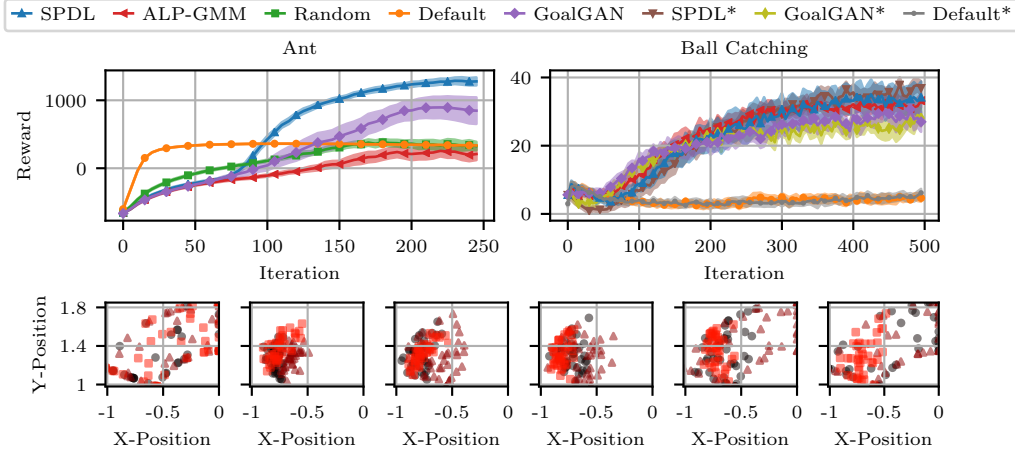

Figure 3: Mean (thick line) and two times standard error (shaded area) of the reward achieved with different curricula in the Ant environment for PPO and in the Ball-Catching environment for SAC (upper plots). The statistics are computed from 20 seeds. For Ball-Catching, runs of SPDL/GoalGAN with an initialized context distribution and runs of Default learning without policy initialization are indicated by asterisks. The lower plots show ball positions in the 'catching' plane sampled from the context distributions $p(c)$ in the Ball-Catching environment at iterations 0, 50, 80, 110, 150 and 200 (from left to right). Different sample colors and shapes indicate different algorithm runs. Given that $p(c)$ is initialized with $\mu(c)$, the samples in iteration 0 visualize the target distribution.

## 5.3 Ball-Catching Environment

Due to a sparse reward function and a broad target task distribution, this final environment is drastically different from the previous ones. In this environment, the agent needs to control a Barrett WAM robot to catch a ball thrown towards it. The reward function is sparse, only rewarding the robot when it catches the ball and penalizing it for excessive movements. In the simulated environment, the ball is said to be caught if it is in contact with the end effector that is attached to the robot. The context $c \in \mathbb{R}^3$ parameterizes the distance to the robot from which the ball is thrown as well as its target position in a plane that intersects the base of the robot. Figure 1 shows the robot as well as the target distribution over the ball positions in the aforementioned 'catching' plane. In this environment, the context $c$ is not visible to the policy, as it only changes the initial state distribution $p(s_0)$ via the encoded target position and initial distance to the robot. Given that the initial state is already observed by the policy, observing the context is superfluous. To tackle this learning task with a curriculum, we initialize the policy of the RL algorithms to hold the robot's initial position. This creates a subspace in the context space in which the policy already performs well, i.e. where the target position of the ball coincides with the initial end effector position. This can be leveraged by CL algorithms.

Since SPDL and GoalGAN support to specify the initial context distribution, we investigate whether this feature can be exploited by choosing the initial context distribution to encode the aforementioned tasks in which the initial policy performs well. When directly learning on the target context distribution without a curriculum, it is not clear whether the policy initialization benefits learning. Hence, we evaluate the performance both with and without a pre-trained policy when not using a curriculum.

Figure 3 and Table 1 show the performance of the investigated curriculum learning approaches. We see that sampling tasks directly from the target distribution does not allow the agent to learn a meaningful policy, regardless of the initial one. Further, all curricula enable learning in this environment and achieve a similar reward. The results also highlight that initialization of the context distribution does not significantly change the performance in this task. The context distributions $p(c)$ visualized in Figure 3 indicate that SPDL shrinks the initially wide context distribution in early iterations to recover the subspace of ball target positions, in which the initial policy performs well. From there, the context distribution then gradually matches the target one. As in the point mass experiment, this happens with differing pace, as can be seen in the visualizations of $p(c)$ in Figure 3 for iteration 200: Two of the three distributions fully match the target distribution while the third only covers half of it.

# 6  Conclusion

We proposed self-paced deep reinforcement learning, an inference-derived curriculum reinforcement learning algorithm. The resulting method is easy to use, allows to draw connections to established regularization techniques for inference, and generalizes previous results in the domain of CRL. In our experiments, the method matched or surpassed performance of other CRL algorithms, especially excelling in tasks where learning is aimed at a single target task.

As discussed, the inference view provides many possibilities for future improvements of the proposed algorithm, such as using more elaborate methods for choosing the hyperparameter $\alpha$ or approximating the variational distribution $q(\boldsymbol{c})$ using more advanced methods. Such algorithmic improvements are expected to further improve the efficiency of the algorithm. Furthermore, a re-interpretation of the self-paced learning algorithm for supervised learning tasks using the presented inference perspective may allow for a unifying view across the boundary of both supervised- and reinforcement learning, allowing to share algorithmic advances.

## Broader Impact

This work proposed a method to speed up and stabilize the learning of autonomous agents via curriculum reinforcement learning. In a practical scenario, such methods can reduce the amount of time, energy, or manual labor required to create autonomous agents for a given task, allowing for economic benefits. Given the inherent goal of RL to create versatile learning algorithms, free of ties to a specific domain, RL algorithms can be used in a variety of fields, ranging from automating aspects of elderly care over autonomous vehicles to military uses. Given the abstract nature of our work, it is, however, hard to estimate the immediate consequences of our work on society, since the algorithmic benefits arising from our work apply equally to all of the aforementioned examples.

## Acknowledgments and Disclosure of Funding

This project has received funding from the DFG project PA3179/1-1 (ROBOLEAP). During parts of the work on this paper, Joni Pajarinen was affiliated with Tampere University, Finland.

## Footnotes

[1] Code for running the experiments can be found at `https://github.com/psclklnk/spdl`

[2] We use the Nvidia Isaac Gym simulator [54] for this experiment.

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
