[Supplementary Material · appendix.pdf]



Figure 1: The extended graphical model used for the presented CRL algorithm. Solid lines mark connections that are present in the 'single task' RL problem. The dashed lines represent the additional connections that occur in the contextual RL setting, where a contextual variable $c$ influences the MDP. Note that $c$ and $\mathcal{O}$ refer to the same variables across all timesteps.

## A  Proofs

We start by restating the Latent-Variable Model for the Contextual RL setting

$$p_{\boldsymbol{\nu},\boldsymbol{\omega}}(\mathcal{O}) = \int p_{\boldsymbol{\nu},\boldsymbol{\omega}}(\mathcal{O},\tau,\boldsymbol{c})d\tau d\boldsymbol{c} \propto \int f(R(\tau,\boldsymbol{c}))p_{\boldsymbol{\omega}}(\tau|\boldsymbol{c})p_{\boldsymbol{\nu}}(\boldsymbol{c})d\tau d\boldsymbol{c}, \tag{1}$$

where $p(\mathcal{O}|\tau,\boldsymbol{c}) \propto f(R(\tau,\boldsymbol{c}))$ with $R(\tau,\boldsymbol{c}) = \sum_{t\geq 0} r_{\boldsymbol{c}}(\boldsymbol{s}_t,\boldsymbol{a}_t)$ and the monotonic transformation $f : \mathbb{R} \mapsto \mathbb{R}_{\geq 0}$ defines the probability of trajectory $\tau$ being optimal in context $\boldsymbol{c}$. In LVM (1), $p_{\boldsymbol{\nu}}(\boldsymbol{c})$ is the distribution over contexts and $p_{\boldsymbol{\omega}}(\tau|\boldsymbol{c})$ is the probability of a trajectory, that depends on the policy $\pi_{\boldsymbol{\omega}}(\boldsymbol{a}|\boldsymbol{s},\boldsymbol{c})$

$$p_{\boldsymbol{\omega}}(\tau|\boldsymbol{c}) = p_{0,\boldsymbol{c}}(\boldsymbol{s}_0)\prod_{t\geq 0} \bar{p}_{\boldsymbol{c}}(\boldsymbol{s}_{t+1}|\boldsymbol{s}_t,\boldsymbol{a}_t)\pi_{\boldsymbol{\omega}}(\boldsymbol{a}_t|\boldsymbol{s}_t,\boldsymbol{c}). \tag{2}$$

Please note the distribution $\bar{p}_{\boldsymbol{c}}(\boldsymbol{s}_{t+1}|\boldsymbol{s}_t,\boldsymbol{a}_t)$. This modified version of the original transition dynamics $p_{\boldsymbol{c}}(\boldsymbol{s}_{t+1}|\boldsymbol{s}_t,\boldsymbol{a}_t)$ is used to account for the discouting factor $\gamma \leq 1$ that is present in the infinite horizon MDP setting. The dynamics $\bar{p}_{\boldsymbol{c}}$ are defined by introducing a terminal state $\boldsymbol{s}_T$, with $r(\boldsymbol{s}_T,\boldsymbol{a}) = 0$ for all $\boldsymbol{a} \in \mathcal{A}$, to which a transition can occur from any state with probability $1 - \gamma$

$$\bar{p}_{\boldsymbol{c}}(\boldsymbol{s}_{t+1}|\boldsymbol{s}_t,\boldsymbol{a}_t) = \begin{cases} 1, & \text{if } \boldsymbol{s}_t = \boldsymbol{s}_T \text{ and } \boldsymbol{s}_{t+1} = \boldsymbol{s}_T \\ 0, & \text{if } \boldsymbol{s}_t = \boldsymbol{s}_T \text{ and } \boldsymbol{s}_{t+1} \neq \boldsymbol{s}_T \\ (1-\gamma), & \text{if } \boldsymbol{s}_t \neq \boldsymbol{s}_T \text{ and } \boldsymbol{s}_{t+1} = \boldsymbol{s}_T \\ \gamma p_{\boldsymbol{c}}(\boldsymbol{s}_{t+1}|\boldsymbol{s}_t,\boldsymbol{a}_t), & \text{else.} \end{cases}$$

Figure 1 visualizes the structure of LVM (1). The term Latent-Variable Model arises because, conceptually, we think about states, actions and contexts as being 'hidden'. This means that there is an underlying distribution over states, actions and contexts, which is, however, marginalized out, leaving only the quantity of interest - the 'optimality' event $\mathcal{O}$. This marginalization makes direct optimization of likelihood (1) intractable. The EM algorithm [1], as applied in the main paper, introduces a variational distribution $q(\boldsymbol{c})$ which decomposes the logarithm of likelihood (1)

$$\log\left(p_{\boldsymbol{\nu},\boldsymbol{\omega}}(\mathcal{O})\right) = \int q(\boldsymbol{c})\log\left(p_{\boldsymbol{\nu},\boldsymbol{\omega}}(\mathcal{O})\right)d\boldsymbol{c} \tag{3}$$

$$= \int q(\boldsymbol{c})\log\left(\frac{q(\boldsymbol{c})}{q(\boldsymbol{c})}\frac{p_{\boldsymbol{\nu},\boldsymbol{\omega}}(\mathcal{O},\boldsymbol{c})}{p_{\boldsymbol{\nu},\boldsymbol{\omega}}(\boldsymbol{c}|\mathcal{O})}\right)d\boldsymbol{c} \tag{4}$$

$$= E_{q(\boldsymbol{c})}\left[\log\left(\frac{p_{\boldsymbol{\nu},\boldsymbol{\omega}}(\mathcal{O},\boldsymbol{c})}{q(\boldsymbol{c})}\right)\right] + D_{\text{KL}}\left(q(\boldsymbol{c})\|p_{\boldsymbol{\nu},\boldsymbol{\omega}}(\boldsymbol{c}|\mathcal{O})\right). \tag{5}$$

The reformulation of the marginal likelihood $p_{\boldsymbol{\nu},\boldsymbol{\omega}}(\mathcal{O})$ between lines (3) and (4) is possible because $p_{\boldsymbol{\nu},\boldsymbol{\omega}}(\mathcal{O},\boldsymbol{c}) = p_{\boldsymbol{\nu},\boldsymbol{\omega}}(\boldsymbol{c}|\mathcal{O})p_{\boldsymbol{\nu},\boldsymbol{\omega}}(\mathcal{O})$. Decomposing the likelihood is beneficial, since it allows to split the optimization of $\log\left(p_{\boldsymbol{\nu},\boldsymbol{\omega}}(\mathcal{O})\right)$ into two steps that can be tackled individually, the so called E- and M-Step. The E-Step minimizes the second term in Eq. (5), yielding $q(\boldsymbol{c}) = p_{\boldsymbol{\nu},\boldsymbol{\omega}}(\boldsymbol{c}|\mathcal{O})$ if $q(\boldsymbol{c})$ is not restricted to a parametric form. The M-Step then maximizes the first term of Eq. (5) w.r.t. $\boldsymbol{\nu}$.

Before proving our first result from the main paper, we quickly note that for our model, the M-Step can be equally thought of as minimizing $D_{\text{KL}}\left(q(\boldsymbol{c})\|p_{\boldsymbol{\nu}}(\boldsymbol{c})\right)$ w.r.t. $\boldsymbol{\nu}$, since

$$E_{q(\boldsymbol{c})}\left[\log\left(\frac{p_{\boldsymbol{\nu},\boldsymbol{\omega}}(\mathcal{O},\boldsymbol{c})}{q(\boldsymbol{c})}\right)\right] = E_{q(\boldsymbol{c})}\left[\log\left(p_{\boldsymbol{\omega}}(\mathcal{O}|\boldsymbol{c})\right)\right] - E_{q(\boldsymbol{c})}\left[\log\left(\frac{q(\boldsymbol{c})}{p_{\boldsymbol{\nu}}(\boldsymbol{c})}\right)\right], \tag{6}$$

where the first term is constant w.r.t. $\boldsymbol{\nu}$ and the second term is equal to $-D_{\text{KL}}\left(q(\boldsymbol{c})\|p_{\boldsymbol{\nu}}(\boldsymbol{c})\right)$.

## A.1 Theorem 1

This theorem establishes the connection between the maximization of our proposed objective for Curriculum generation w.r.t. $\boldsymbol{\nu}$

$$\max_{\boldsymbol{\nu}} J(\boldsymbol{\nu}, \boldsymbol{\omega}) - \alpha D_{\mathrm{KL}}\left(p_{\boldsymbol{\nu}}(\boldsymbol{c}) \| \mu(\boldsymbol{c})\right), \quad \alpha \geq 0 \tag{7}$$

and the discussed EM algorithm when applied to LVM 1. More precisely, we show that modifications of the E-Step allow to relate Objective (7) to the execution of the E- and M-Step.

**Theorem 1.** *Choosing $f(\cdot) = \exp(\cdot)$, maximizing Objective (7) minus a KL divergence term $D_{KL}\left(p_{\boldsymbol{\nu}}(\boldsymbol{c}) \| p_{\tilde{\boldsymbol{\nu}}}(\boldsymbol{c})\right)$ is equal to executing E- and M-Step while restricting $q(\boldsymbol{c})$ to be of the same parametric form as $p_{\boldsymbol{\nu}}(\boldsymbol{c})$ and introducing a regularized E-Step $D_{KL}\left(q(\boldsymbol{c}) \middle\| \frac{1}{Z} p_{\tilde{\boldsymbol{\nu}}, \boldsymbol{\omega}}(\boldsymbol{c}|\mathcal{O})^{\frac{1}{1+\alpha}} \mu(\boldsymbol{c})^{\frac{\alpha}{1+\alpha}}\right)$.*

*Proof.* As we restrict $q(\boldsymbol{c})$ to be of the same parametric form as $p_{\boldsymbol{\nu}}(\boldsymbol{c})$, an M-Step becomes superfluous, because the optimal solution of this M-Step clearly matches $q(\boldsymbol{c})$. We see that, when restricting $q(\boldsymbol{c})$ to the same parametric form as $p_{\boldsymbol{\nu}}(\boldsymbol{c})$, executing E- and M-Step is equal to simply minimzing the E-Step, where $q(\boldsymbol{c})$ is replaced by $p_{\boldsymbol{\nu}}(\boldsymbol{c})$

$$\min_{\boldsymbol{\nu}} D_{\mathrm{KL}}\left(p_{\boldsymbol{\nu}}(\boldsymbol{c}) \middle\| \frac{1}{Z} p_{\tilde{\boldsymbol{\nu}}, \boldsymbol{\omega}}(\boldsymbol{c}|\mathcal{O})^{\frac{1}{1+\alpha}} \mu(\boldsymbol{c})^{\frac{\alpha}{1+\alpha}}\right). \tag{8}$$

Consequently, we are left to show that above optimization problem is the same as the maximization of Objective 7. This is, however, a task of simple reformulation

$$\min_{\boldsymbol{\nu}} D_{\mathrm{KL}}\left(p_{\boldsymbol{\nu}}(\boldsymbol{c}) \middle\| \frac{1}{Z} p_{\tilde{\boldsymbol{\nu}}, \boldsymbol{\omega}}(\boldsymbol{c}|\mathcal{O})^{\frac{1}{1+\alpha}} \mu(\boldsymbol{c})^{\frac{\alpha}{1+\alpha}}\right) \tag{9}$$

$$= \min_{\boldsymbol{\nu}} Z + E_{p_{\boldsymbol{\nu}}(\boldsymbol{c})}\left[\log\left(\frac{p_{\boldsymbol{\nu}}(\boldsymbol{c})}{p_{\tilde{\boldsymbol{\nu}}, \boldsymbol{\omega}}(\boldsymbol{c}|\mathcal{O})^{\frac{1}{1+\alpha}} \mu(\boldsymbol{c})^{\frac{\alpha}{1+\alpha}}}\right)\right] \tag{10}$$

$$= \max_{\boldsymbol{\nu}} -Z + \frac{1}{1+\alpha} E_{p_{\boldsymbol{\nu}}(\boldsymbol{c})}\left[\log\left(p_{\boldsymbol{\omega}}(\mathcal{O}|\boldsymbol{c})\right)\right] + \frac{1}{1+\alpha} p_{\tilde{\boldsymbol{\nu}}, \boldsymbol{\omega}}(\mathcal{O}) \tag{11}$$

$$- E_{p_{\boldsymbol{\nu}}(\boldsymbol{c})}\left[\log\left(\frac{p_{\boldsymbol{\nu}}(\boldsymbol{c})}{p_{\tilde{\boldsymbol{\nu}}}(\boldsymbol{c})^{\frac{1}{1+\alpha}} \mu(\boldsymbol{c})^{\frac{\alpha}{1+\alpha}}}\right)\right]. \tag{12}$$

Before proceeding to reformulate above KL-Divergence, we note that we can simply remove the normalization constant $Z$ as well as the term $\frac{1}{1+\alpha} p_{\tilde{\boldsymbol{\nu}}, \boldsymbol{\omega}}(\mathcal{O})$, since they are constant w.r.t. $\boldsymbol{\nu}$. Furthermore, we can rescale the objective by $1 + \alpha$ without changing the optimal solution, yielding

$$\max_{\boldsymbol{\nu}} E_{p_{\boldsymbol{\nu}}(\boldsymbol{c})}\left[\log\left(p_{\boldsymbol{\omega}}(\mathcal{O}|\boldsymbol{c})\right)\right] - (1+\alpha) E_{p_{\boldsymbol{\nu}}(\boldsymbol{c})}\left[\log\left(\frac{p_{\boldsymbol{\nu}}(\boldsymbol{c})}{p_{\tilde{\boldsymbol{\nu}}}(\boldsymbol{c})^{\frac{1}{1+\alpha}} \mu(\boldsymbol{c})^{\frac{\alpha}{1+\alpha}}}\right)\right] \tag{13}$$

$$= \max_{\boldsymbol{\nu}} E_{p_{\boldsymbol{\nu}}(\boldsymbol{c})}\left[\log\left(p_{\boldsymbol{\omega}}(\mathcal{O}|\boldsymbol{c})\right)\right] - D_{\mathrm{KL}}\left(p_{\boldsymbol{\nu}}(\boldsymbol{c}) \| p_{\tilde{\boldsymbol{\nu}}}(\boldsymbol{c})\right) - \alpha D_{\mathrm{KL}}\left(p_{\boldsymbol{\nu}}(\boldsymbol{c}) \| \mu(\boldsymbol{c})\right). \tag{14}$$

The last reformulation was possible since we can write $p_{\boldsymbol{\nu}}(\boldsymbol{c}) = p_{\boldsymbol{\nu}}(\boldsymbol{c})^{\frac{1}{1+\alpha}} p_{\boldsymbol{\nu}}(\boldsymbol{c})^{\frac{\alpha}{1+\alpha}}$. To proof Theorem 1, we simply need to relate the quantity $E_{p_{\boldsymbol{\nu}}(\boldsymbol{c})}\left[\log\left(p_{\boldsymbol{\omega}}(\mathcal{O}|\boldsymbol{c})\right)\right]$ to $J(\boldsymbol{\nu}, \boldsymbol{\omega})$. Using $f(R(\tau, \boldsymbol{c})) = \exp(R(\tau, \boldsymbol{c}))$ and Jensens inequality, we can show that

$$\log\left(p_{\boldsymbol{\omega}}(\mathcal{O}|\boldsymbol{c})\right) = \log\left(\int \exp(R(\tau, \boldsymbol{c})) p_{\boldsymbol{\omega}}(\tau|\boldsymbol{c}) d\tau\right) - \log(Z) \tag{15}$$

$$\geq \int R(\tau, \boldsymbol{c}) p_{\boldsymbol{\omega}}(\tau|\boldsymbol{c}) d\tau - \log(Z) \tag{16}$$

$$= \int \sum_{t \geq 0} r(\boldsymbol{s}_t, \boldsymbol{a}_t) p_{0, \boldsymbol{c}}(\boldsymbol{s}_0) \prod_{t \geq 0} \bar{p}_{\boldsymbol{c}}(\boldsymbol{s}_{t+1}|\boldsymbol{s}_t, \boldsymbol{a}_t) \pi_{\boldsymbol{\omega}}(\boldsymbol{a}_t|\boldsymbol{s}_t, \boldsymbol{c}) d\boldsymbol{s}_t d\boldsymbol{a}_t - \log(Z) \tag{17}$$

$$= \int \sum_{t \geq 0} \gamma^t r(\boldsymbol{s}_t, \boldsymbol{a}_t) p_{0, \boldsymbol{c}}(\boldsymbol{s}_0) \prod_{t \geq 0} p_{\boldsymbol{c}}(\boldsymbol{s}_{t+1}|\boldsymbol{s}_t, \boldsymbol{a}_t) \pi_{\boldsymbol{\omega}}(\boldsymbol{a}_t|\boldsymbol{s}_t, \boldsymbol{c}) d\boldsymbol{s}_t d\boldsymbol{a}_t - \log(Z)$$
$$\tag{18}$$

$$= E_{p_{0, \boldsymbol{c}}(\boldsymbol{s})}\left[V_{\boldsymbol{\omega}}(\boldsymbol{s}, \boldsymbol{c})\right] - \log(Z) \tag{19}$$

The reformulation between lines (17) and (18) is possible because of the modified dynamics. The chance of not transitioning into $s_T$ for $t$ steps is given by $\gamma^t$. Since the agent recieves no reward in $s_T$, any terms of the form $r(s_T, a_t)$ can be removed from the expectation in line (17). Combining these two observations yields line (18). Given that the normalization constant $Z$ is constant across all contexts $c$, we can again remove it from the optimization of the reformulated E-Step (Eq. 14). With that we see that when choosing $f(R(\tau, c)) = \exp(R(\tau, c))$, it holds that $E_{p_\nu(c)}[\log(p_\omega(\mathcal{O}|c))] \geq J(\nu, \omega)$. Consequently, we optimize the E-Step using a lower bound by optimizing Objective 7. Given that we can skip the M-Step due to restricting the form of $q(c)$, we see that we are indeed performing the steps of the EM algorithm outlined in Theorem 1. $\qquad\square$

### A.2 Theorem 2

This theorem shows that the update rule for the context distribution, established by Klink et al. [2], is also explained as applying EM to maximize $p_{\nu,\omega}(\mathcal{O})$ w.r.t. $\nu$. In this case, however, the variational distribution is not restricted to a parametric form, requiring an explicit M-Step. Looking at the work by Klink et al. [2], we see that their algorithm indeed performs an M-Step by fitting a parametric model to weighted samples (which approximately represent $q(c)$).

**Theorem 2.** *Choosing $f(\cdot) = \exp(\cdot/\eta)$, the (unrestricted) variational distribution after the regularized E-Step is given by $q(c) \propto p_\nu(c) \exp\left(\frac{V_\omega(c) + \eta\alpha(\log(\mu(c)) - \log(p_\nu(c)))}{\eta + \eta\alpha}\right)$, where $V_\omega(c)$ is the 'episodic value function' as defined in [3].*

*Proof.* We first note that, given that we are not restricting $q(c)$ to any parametric form, $q(c) = \frac{1}{Z} p_{\tilde{\nu},\omega}(c|\mathcal{O})^{\frac{1}{1+\alpha}} \mu(c)^{\frac{\alpha}{1+\alpha}}$ holds after the E-Step. A reformulation of this probabilitty distribution brings us closer to the desired result

$$\frac{1}{Z} p_{\tilde{\nu},\omega}(c|\mathcal{O})^{\frac{1}{1+\alpha}} \mu(c)^{\frac{\alpha}{1+\alpha}} \tag{20}$$

$$\propto p_\omega(\mathcal{O}|c)^{\frac{1}{1+\alpha}} p_\nu(c)^{\frac{1}{1+\alpha}} \mu(c)^{\frac{\alpha}{1+\alpha}} \tag{21}$$

$$\propto p_\nu(c) p_\omega(\mathcal{O}|c)^{\frac{1}{1+\alpha}} \mu(c)^{\frac{\alpha}{1+\alpha}} p_\nu(c)^{\frac{-\alpha}{1+\alpha}} \tag{22}$$

$$\propto p_\nu(c) \exp\left(\log\left(p_\omega(\mathcal{O}|c)^{\frac{1}{1+\alpha}} \mu(c)^{\frac{\alpha}{1+\alpha}} p_\nu(c)^{\frac{-\alpha}{1+\alpha}}\right)\right) \tag{23}$$

$$\propto p_\nu(c) \exp\left(\frac{\log(p_\omega(\mathcal{O}|c)) + \alpha(\log(\mu(c)) - \log(p_\nu(c)))}{1+\alpha}\right) \tag{24}$$

To proof Theorem 2, we need to relate $\log(p_\omega(\mathcal{O}|c))$ to the 'episodic value function' $V_\omega(c) = \eta\log\left(\int \exp(R(\tau|c)/\eta) p_\omega(\tau|c)d\tau\right)$ as defined in [3]. By choosing the transformation $f(R(\tau, c)) = \exp\left(\frac{R(\tau,c)}{\eta}\right)$, it follows that

$$\log(p_\omega(\mathcal{O}|c)) \tag{25}$$

$$= \log\left(\int p(\mathcal{O}|\tau, c) p_\omega(\tau|c)d\tau\right) \tag{26}$$

$$\propto \log\left(\int \exp\left(\frac{R(\tau, c)}{\eta}\right) p_\omega(\tau|c)d\tau\right) = \frac{1}{\eta}V_\omega(c). \tag{27}$$

Inserting this result into Eq. (24) proofs the theorem. $\qquad\square$

## B Experimental Details

In this section, we present details that could not be included in the main paper due to space limitations. This includes parameters of the employed algorithms, additional details about the mechanics of the environments as well as a qualitative discussion of the results.

The parameters of SPDL for different environments and RL algorithms are shown in Table 1. The parameters $N_\alpha$ and $\zeta$ have the same meaning as in the main paper. The additional parameter $n_{\text{OFFSET}}$ describes the number of RL algorithm iterations that take place before SPDL is allowed the change

Table 1: Hyperparameters for the SPDL algorithm per environment and RL algorithm. The asterisks in the table mark the Ball-Catching experiments with an initialized context distribution.

| | $N_\alpha$ | $\zeta$ | $n_{\text{OFFSET}}$ | $n_{\text{STEP}}$ | $\boldsymbol{\sigma}_{\text{LB}}$ | $D_{\text{KL}_{LB}}$ |
|---|---|---|---|---|---|---|
| POINT-MASS (TRPO) | 70 | 1.6 | 5 | 2048 | [0.2  0.1875  0.1] | 8000 |
| POINT-MASS (PPO) | 10 | 1.4 | 5 | 2048 | [0.2  0.1875  0.1] | 8000 |
| POINT-MASS (SAC) | 50 | 1.2 | 5 | 2048 | [0.2  0.1875  0.1] | 8000 |
| ANT (PPO) | 15 | 0.4 | 40 | 81920 | [1  0.5] | 11000 |
| BALL-CATCHING (TRPO) | 70 | 0.4 | 5 | 5000 | - | - |
| BALL-CATCHING* (TRPO) | 0 | 0.425 | 5 | 5000 | - | - |
| BALL-CATCHING (PPO) | 50 | 0.45 | 5 | 5000 | - | - |
| BALL-CATCHING* (PPO) | 0 | 0.45 | 5 | 5000 | - | - |
| BALL-CATCHING (SAC) | 60 | 0.6 | 5 | 5000 | - | - |
| BALL-CATCHING* (SAC) | 0 | 0.6 | 5 | 5000 | - | - |

the context distribution. This parameter can be necessary if some iterations are required until the approximated value function produces meaningful estimates of the expected value. In the ant environment, we realized that the agent takes a certain amount of time (roughly 40 iterations) until it manages to reach the wall. Only then, the difference in task difficulty becomes apparent. The parameter $n_{\text{OFFSET}}$ allows to compensate for such task-specific details. This procedure corresponds to providing parameters of a pre-trained policy as $\boldsymbol{\omega}_0$ in the algorithm sketched in the main paper. We selected the best $\zeta$ for every RL algorithm by a simple grid-search in an interval around a reasonably working parameter that was found by simple trial and error. For the PointMass environment, we only tuned the hyperparameters for SPDL in the experiment with a three-dimensional context space and reused them for the two-dimensional context space. To conduct the experiments, we use the implementation of ALP-GMM, GoalGAN and SPRL provided in the repositories accompanying the papers [4, 5, 2].

For ALP-GMM we tuned the percentage of random samples drawn from the context space $p_{\text{RAND}}$, the number of policy rollouts between the update of the context distribution $n_{\text{ROLLOUT}}$ as well as the maximum buffer size of past trajectories to keep $s_{\text{BUFFER}}$. For each environment and algorithm, we did a grid-search over

$$(p_{\text{RAND}}, n_{\text{ROLLOUT}}, s_{\text{BUFFER}}) \in \{0.1, 0.2, 0.3\} \times \{50, 100, 200\} \times \{500, 1000, 2000\}.$$

For GoalGAN we tuned the amount of random noise that is added on top of each sample $\delta_{\text{NOISE}}$, the number of policy rollouts between the update of the context distribution $n_{\text{ROLLOUT}}$ as well as the percentage of samples drawn from the success buffer $p_{\text{SUCCESS}}$. For each environment and algorithm, we did a grid-search over

$$(\delta_{\text{NOISE}}, n_{\text{ROLLOUT}}, p_{\text{SUCCESS}}) \in \{0.025, 0.05, 0.1\} \times \{50, 100, 200\} \times \{0.1, 0.2, 0.3\}.$$

The results of the hyperparameter optimization for GoalGAN and ALP-GMM are shown in Table 2.

The similarity of our algorithm and SPRL – and since we could only apply it to one experiment due to numerical reasons – allowed to start from the parameters of SPDL and obtain well-working parameters by a few adjustments.

In the experiments, we found that restricting the standard deviation of the context distribution to stay above a certain lower bound $\boldsymbol{\sigma}_{\text{LB}}$ helps to stabilize learning when generating curricula for narrow target distributions with SPDL. Although such constraints could be included rigorously via constraints on the distribution $p_{\boldsymbol{\nu}}(\boldsymbol{c})$ in the E-Step, we accomplish this by just clipping the standard deviation until the KL-Divergence w.r.t. the target distribution falls below a certain threshold $D_{\text{KL}_{\text{LB}}}$. This technique was also employed by Klink et al. [2].

Opposed to the sketched algorithm in the main paper, we specify the number of steps $n_{\text{STEP}}$ in the environment instead of the number of trajectory rollouts between context distribution updates in our implementation.

Since for all environments, both initial- and target distribution are Gaussians with independent noise in each dimension, we specify them in Table 3 by providing their mean $\boldsymbol{\mu}$ and the vector of standard deviations for each dimension $\boldsymbol{\delta}$. When sampling from a Gaussian, the resulting context is clipped to stay in the defined context space.

Table 2: Hyperparameters for the ALP-GMM and GoalGAN algorithm per environment and RL algorithm. The abbreviation AG is used for ALP-GMM, while GG stands for GoalGAN.

| | $p_{\text{RAND}}$ | $n_{\text{ROLLOUT}_{\text{AG}}}$ | $s_{\text{BUFFER}}$ | $\delta_{\text{NOISE}}$ | $n_{\text{ROLLOUT}_{\text{GG}}}$ | $p_{\text{SUCCESS}}$ |
|---|---|---|---|---|---|---|
| POINT-MASS 3D (TRPO) | 0.1 | 100 | 1000 | 0.05 | 200 | 0.2 |
| POINT-MASS 3D (PPO) | 0.1 | 100 | 500 | 0.025 | 200 | 0.1 |
| POINT-MASS 3D (SAC) | 0.1 | 200 | 1000 | 0.1 | 100 | 0.1 |
| POINT-MASS 2D (TRPO) | 0.3 | 100 | 500 | 0.1 | 200 | 0.2 |
| POINT-MASS 2D (PPO) | 0.2 | 100 | 500 | 0.1 | 200 | 0.3 |
| POINT-MASS 2D (SAC) | 0.2 | 200 | 1000 | 0.025 | 50 | 0.2 |
| ANT (PPO) | 0.1 | 50 | 500 | 0.05 | 125 | 0.2 |
| BALL-CATCHING (TRPO) | 0.2 | 200 | 2000 | 0.1 | 200 | 0.3 |
| BALL-CATCHING (PPO) | 0.3 | 200 | 2000 | 0.1 | 200 | 0.3 |
| BALL-CATCHING (SAC) | 0.3 | 200 | 1000 | 0.1 | 200 | 0.3 |

If necessary, we tuned the hyperparameters of the RL algorithms by hand on easier versions of the target task, not employing any Curriculum. The goal was to be as fair as possible by not optimizing the RL algorithm for a specific curriculum. For the Ant and PointMass environment, this was done by training on a wide gate positioned right in front of the agent. For the Ball-Catching environment, this was done by training on a version of the environment with dense reward. For PPO, we use the "PPO2" implementation of Stable-Baselines.

The experiments were conducted on a computer with an AMD Ryzen 9 3900X 12-Core Processor, an Nvidia RTX 2080 graphics card and 64GB of RAM.

## B.1 Point-Mass Environment

The state of this environment is comprised of the position and velocity of the point-mass $\boldsymbol{s} = [x\ \dot{x}\ y\ \dot{y}]$. The actions correspond to the force applied in x- and y-dimension $\boldsymbol{a} = [F_x\ F_y]$. The context encodes position and width of the gate as well as the dynamic friction coefficient of the ground on which the point mass slides $\boldsymbol{c} = [p_g\ w_g\ \mu_k] \in [-4, 4] \times [0.5, 8] \times [0, 4] \subset \mathbb{R}^3$. The dynamics of the system are defined by

$$\begin{pmatrix} \dot{x} \\ \ddot{x} \\ \dot{y} \\ \ddot{y} \end{pmatrix} = \begin{pmatrix} 0 & 1 & 0 & 0 \\ 0 & -\mu_k & 0 & 0 \\ 0 & 0 & 0 & 1 \\ 0 & 0 & 0 & -\mu_k \end{pmatrix} \boldsymbol{s} + \begin{pmatrix} 0 & 0 \\ 1 & 0 \\ 0 & 0 \\ 0 & 1 \end{pmatrix} \boldsymbol{a}.$$

The $x$- and $y$- position of the point mass is enforced to stay within the space $[-4, 4] \times [-4, 4]$. The gate is located at position $[p_g\ 0]$. If the agent crosses the line $y = 0$, we check whether its $x$-position is within the interval $[p_g - 0.5w_g, p_g + 0.5w_g]$. If this is not the case, we stop the episode as the agent has crashed into the wall. Each episode is terminated after a maximum of 100 steps. The reward function is given by

$$r(\boldsymbol{s}, \boldsymbol{a}) = \exp\left(-0.6\|\boldsymbol{o} - [x\ y]\|_2\right),$$

where $\boldsymbol{o} = [0\ -3]$, $\|\cdot\|_2$ is the L2-Norm. The agent is always initialized at state $\boldsymbol{s}_0 = [0\ 0\ 3\ 0]$.

For all RL algorithms, we use a discount factor of $\gamma = 0.95$ and represent policy and value function by networks using 21 hidden layers with tanh activations. For TRPO and PPO, we take 2048 steps in the environment between policy updates.

For TRPO we set the GAE parameter $\lambda = 0.99$, the maximum allowed KL-Divergence to $0.004$ and the value function step size $a_v \approx 0.24$, leaving all other parameters to their implementation defaults.

For PPO we use GAE parameter $\lambda = 0.99$, an entropy coefficient of $0$ and disable the clipping of the value function objective. The number of optimization epochs is set to $8$ and we use $32$ mini-batches. All other parameters are left to their implementation defaults.

For SAC, we use an experience-buffer of $10000$ samples, leaving every other setting to the implementation default. Hence we use the soft Q-Updates and update the policy after every environment step.

Figure 2: Visualizations of policy rollouts in the Point-Mass Environment (three context dimensions) with policies learned using different curricula and RL algorithms. Each rollout was generated using a policy learned with a different seed. The first row shows results for TRPO, the second for PPO and the third shows results for SAC.

For SPRL, we use $K_\alpha = 40$, $n_{\text{OFFSET}} = 0$, $\zeta = 2.0$ for the 3D- and $\zeta = 1.5$ and 2D case. We use the same values for $\sigma_{\text{LB}}$ and $D_{\text{KL}_{\text{LB}}}$ as for SPDL (Table 1). Between updates of the episodic policy, we do 25 policy rollouts and keep a buffer containing rollouts from the past 10 iterations, resulting in 250 samples for policy- and context distribution update. The linear policy over network weights is initialized to a zero-mean Gaussian with unit variance. We use Polynomial features up to degree two for approximating the value function during policy optimization. For the allowed KL-Divergence, we observed best results when using $\epsilon = 0.5$ for the weight computation of the samples, but using a lower value of $\epsilon = 0.2$ when fitting the parametric policy to these weighted samples. We suppose that the higher value of $\epsilon$ during weight computation counteracts the effect of the buffer containing policy samples from earlier iterations.

Looking at Figure 2, we can see that ALP-GMM allowed to learn policies that sometimes are able to pass the gate. However, in other cases, the policies crashed the point mass into the wall. Opposed to this, directly training on the target task led to policies that learned to steer the point mass very close to the wall without crashing (which is unfortunately hard to see in the plot). Reinvestigating the above reward function, this explains the lower reward of ALP-GMM, GoalGAN and the randomly generated curriculum compared to directly learning on the target task, as a crash prevents the agent from accumulating positive rewards over time.

## B.2 Ant Environment

As mentioned in the main paper, we simulate the ant using the Isaac Gym simulator [6]. This allows to speed up training time by parallelizing the simulation of policy rollouts on the graphics card. Since the Stable-Baselines implementation of TRPO and SAC do not support the use of vectorized

Figure 3: Visualizations of the $x$-position during policy rollouts in the Ant Environment with policies learned using different curricula. The blue lines correspond to 200 individual trajectories and the thick black line shows the median over these individual trajectories. The trajectories were generated from 20 algorithms runs, were each final policy was used to generate 10 trajectories.

environments, it is hard to combine Isaac Gym with these algorithms. Because of this reason, we decided not to run experiments with TRPO and SAC in the Ant environment.

The state $s \in \mathbb{R}^{29}$ is defined to be the 3D-position of the ant's body, its angular and linear velocity as well as positions and velocities of the 8 joints of the ant. An action $a \in \mathbb{R}^8$ is defined by the 8 torques that are applied to the ant's joints.

The context $c = [p_g \; w_g] \in [-10, 10] \times [3, 13] \subset \mathbb{R}^2$ defines, just as in the Point-Mass environment, the position and width of the gate that the Ant needs to pass.

The reward function of the environment is computed based on the $x$-position of the ant's center of mass $c_x$ in the following way

$$r(s, a) = 1 + 5 \exp \left( -0.5 \min(0, c_x - 4.5)^2 \right) - 0.3 \|a\|_2^2.$$

The constant 1 term was taken from the OpenAI Gym implementation to encourage the survival of the ant [7]. Compared to the OpenAI Gym environment, we set the armature value of the joints from 1 to 0 and also decrease the maximum torque from 150Nm to 20Nm, since the values from OpenAI Gym resulted in unrealistic movement behavior in combination with Isaac Gym. Nonetheless, these changes did not result in a qualitative change in the algorithm performances.

With the wall being located at position $x=3$, the agent needs to pass it in order to obtain the full environment reward by ensuring that $c_x >= 4.5$.

The policy and value function are represented by neural networks with two hidden layers of 64 neurons each and $\tanh$ activation functions. We use a discount factor $\gamma = 0.995$ for all algorithms, which can be explained due to the long time horizons of 750 steps. We take 81920 steps in the environment between a policy update. This was significantly sped-up by the use of the Isaac Gym simulator, which allowed to simulate 40 environments in parallel on a single GPU.

For PPO, we use an entropy coefficient of 0 and disable the clipping of the value function objective. All other parameters are left to their implementation defaults. We disable the entropy coefficient as we observed that for the Ant environment, PPO still tends to keep around $10 - 15\%$ of its initial additive noise even during late iterations.

Investigating Figure 3, we see that both SPDL and GoalGAN learn policies that allow to pass the gate. However, the policies learned with SPDL seem to be more reliable compared to the ones learned with GoalGAN. As mentioned in the main paper, ALP-GMM and a random curriculum also

learn policies that navigate the ant towards the goal in order to pass it. However, the behavior is less directed and less reliable. Interestingly, directly learning on the target task results in a policy that tends to not move in order to avoid action penalties. Looking at the main paper, we see that this results in a similar reward compared to the inefficient policies learned with ALP-GMM and a random curriculum.

## B.3 Ball-Catching Environment

In the final environment, the robot is controlled in joint space via the desired position for 5 of the 7 joints. We only control a subspace of all available joints, since it is not necessary for the robot to leave the "catching" plane (defined by $x = 0$) that is intersected by each ball. The actions $\boldsymbol{a} \in \mathbb{R}^5$ are defined as the displacement of the current desired joint position. The state $\boldsymbol{s} \in \mathbb{R}^{21}$ consists of the positions and velocities of the controlled joints, their current desired positions, the current three-dimensional ball position and its linear velocity.

As previously mentioned, the reward function is sparse,

$$r(\boldsymbol{s}, \boldsymbol{a}) = 0.275 - 0.005\|\boldsymbol{a}\|_2^2 + \begin{cases} 50 + 25(\boldsymbol{n}_s \cdot \boldsymbol{v}_b)^5, & \text{if ball caught} \\ 0, & \text{else} \end{cases},$$

only giving a meaningful reward when catching the ball and otherwise just a slight penalty on the actions to avoid unnecessary movements. In the above definition, $\boldsymbol{n}_s$ is a normal vector of the end effector surface and $\boldsymbol{v}_b$ is the linear velocity of the ball. This additional term is used to encourage the robot to align its end effector with the curve of the ball. If the end effector is e.g. a net (as assumed for our experiment), the normal is chosen such that aligning it with the ball maximizes the opening through which the ball can enter the net.

The context $c = [\phi, r, d_x] \in [0.125\pi, 0.5\pi] \times [0.6, 1.1] \times [0.75, 4] \subset \mathbb{R}^3$ controls the target ball position in the catching plane, i.e.

$$\boldsymbol{p}_{\text{des}} = \begin{bmatrix} 0 & -r\cos(\phi) & 0.75 + r\sin(\phi) \end{bmatrix}.$$

Furthermore, the context determines the distance in $x$-dimension from which the ball is thrown

$$\boldsymbol{p}_{\text{init}} = [d_x \ d_y \ d_z],$$

where $d_y \sim \mathcal{U}(-0.75, -0.65)$ and $d_z \sim \mathcal{U}(0.8, 1.8)$ and $\mathcal{U}$ represents the uniform distribution. The initial velocity is then computed using simple projectile motion formulas by requiring the ball to reach $\boldsymbol{p}_{\text{des}}$ at time $t = 0.5 + 0.05d_x$. As we can see, the context implicitly controls the initial state of the environment.

The policy and value function networks for the RL algorithms have three hidden layers with 64 neurons each and tanh activation functions. We use a discount factor of $\gamma = 0.995$. The policy updates in TRPO and PPO are done after 5000 environment steps.

For SAC, a replay buffer size of $100,000$ is used. Due to the sparsity of the reward, we increase the batch size to 512. Learning with SAC starts after 1000 environment steps. All other parameters are left to their implementation defaults.

For TRPO we set the GAE parameter $\lambda = 0.95$, leaving all other parameters to their implementation defaults.

For PPO we use a GAE parameter $\lambda = 0.95$, 10 optimization epochs, 25 mini-batches per epoch, an entropy coefficient of 0 and disable the clipping of the value function objective. The remaining parameters are left to their implementation defaults.

Table 3: Mean and standard deviation of target and initial distributions per environment.

| | $\boldsymbol{\mu}_{\text{INIT}}$ | $\boldsymbol{\delta}_{\text{INIT}}$ | $\boldsymbol{\mu}_{\text{TARGET}}$ | $\boldsymbol{\delta}_{\text{TARGET}}$ |
|---|---|---|---|---|
| POINT-MASS | [0 4.25 2] | [2 1.875 1] | [2.5 0.5 0] | [0.004 0.00375 0.002] |
| ANT | [0 8] | [3.2 1.6] | [−8 3] | [0.01 0.005] |
| BALL-CATCHING | [0.68 0.9 0.85] | [0.03 0.03 0.3] | [1.06 0.85 2.375] | [0.8 0.38 1] |

Figure 4: Mean Catching Rate of the final policies learned with different curricula and RL algorithms on the Ball Catching environment. The mean is computed from 20 algorithm runs with different seeds. For each run, the success rate is computed from 200 ball-throws. The bars visualize the estimated standard error.

Figure 4 visualizes the catching success rates of the learned policies. As can be seen, the performance of the policies learned with the different RL algorithms achieve comparable catching performance. Interestingly, SAC performs comparable in terms of catching performance, although the average reward of the final policies learned with SAC is lower. This is to be credited to excessive movement and/or bad alignment of the end effector with the velocity vector of the ball.