[Reviews · NeurIPS 2020]

Review 1

Summary and Contributions: # UPDATE After reading authors rebuttal, and discussion with other reviewers, I would like to recommend acceptance of this paper (8). # REVIEW In this paper authors present a new method of Curriculum Reinforcement Learning (CRL) based on a theoretical foundations built from viewing RL as inference and exploiting duality to EM algorithms over latent variable models. Resulting formulation (eq. 3) is very explicit and easy to interpret. Authors show connection to existing methods (e.g. Klink et al.) and explain some of their results as a consequence. Finally, authors propose a tracktable algorithm that approximates the theoretical model, and evaluate it on a range of tasks, with varied domains and underlying RL algorithms, showing broad application of a proposed method.

Strengths: - A clear mathematical intuition behind the problem, with a formulation of curriculum learning as a traditional expectation maximisation, with a training time controllable distribution over context - Grounding method in the RL as inference view, which will allow other researchers to test other potential approximation techniques and thus push results even further - A clear description of the algorithmization of the proposed approach - Evaluation on not only varied environments, but also varied underlying RL algorithms - Providing a connecting/unifying result to a paper of Klink et al.

Weaknesses: - Because of its math heavy presentation the paper might reach smaller audience than it could. The resulting algorithm is relatively simple, and easy to understand, but providing a relatively heavy introduction to it might be discouraging. Reviewer would suggest reorganising it to especially move relation to work of Klink et al. towards the end of the paper, as despite being an interesting result, it is not a required part to understand to appreciate and use, the method proposed. - It is quite unclear how one should choose the penalty proportion \xi, nor how it was selected in the experiments (this information is absent from the main text).

Correctness: To reviewer's best knowledge paper contains correct claims. However, reviewer did not verify correctness of Theorems 1 and 2, and thus lowered the confidence score to level 3.

Clarity: Overall presentation is very clear, all the figures are easy to read, notation consistent, and language used easy to follow. The minor clarity issues are: - N_\alpha is never explained in the text - the notation of p_v vs p_{v_i} is a bit confusing when v is optimisation variable and v_i is its value at i'th iteration, is there a way to make the relation more clear?

Relation to Prior Work: Relation to the previous work is clearly discussed, furthermore one of the main contributions is a nice connection to a work of Klink et al.

Reproducibility: Yes

Additional Feedback:


Review 2

Summary and Contributions: This paper uses the "RL as inference" framework to derive a joint objective over a policy, and a distribution over tasks. The aim is that by learning these two quantities simultaneously, the task distribution can select tasks that provide useful signal at its current stage of learning, before converging to some desired distribution over tasks asymptotically. POST-REBUTTAL I thank the authors for their rebuttal, particularly the clarifications on the assumptions around task distributions. I have increased my rating to a 7.

Strengths: The paper proposes a neat objective in Eqn (3) for simultaneously learning a policy, and distribution over tasks. There are nice connections drawn with the "RL as inference" framework, and the experimental results show that this approach works well on several tasks.

Weaknesses: I would have appreciated more discussion around the generality of this approach relative to other approaches to curriculum learning in RL. In particular, this approach seems to require a parametrisation of context/task space (unlike some of the approaches compared against, such as GoalGAN), and advance knowledge of the target distribution over contexts (in contrast to standard RL, where declarative knowledge of the rewards isn't required up front). This still seems like an interesting contribution to me, but the paper would be stronger if (i) the salience of these assumptions for RL applications was discussed further, and (ii) the discussion around the experiments highlighted these differences between the methods in more detail.

Correctness: The technical content of the paper seems broadly correct. I have checked the proofs of Theorems 1 and 2 and believe there may be some minor issues, but I expect these can be clarified by the authors.

Clarity: Overall I found the paper to be quite clearly written, although I found the discussion of connections to the EM algorithm too brief in the main paper (the authors give a clearer description in the appendix). Given the importance of this connection to the two theorems of the paper, I would encourage the authors to give further details on the connection in the main paper in future versions.

Relation to Prior Work: The paper does a good job of contextualising itself against earlier work in curriculum learning and RL as inference. There is some overlap of the theoretical work with Klink et al. (2019), as the authors highlight (although I believe the connections to EM are new) so the main novelty relative to prior work is in the application of this framework to deep RL.

Reproducibility: Yes

Additional Feedback: Section 4 As mentioned above, the main objective (3) seems to require a suitable parametrisation of contexts/tasks in advance, as well as explicit knowledge of the desired distribution over contexts. This assumption seems quite strong, but isn't discussed in much detail in the paper. Can the authors comment on whether the method is applicable when a low-dimensional context parametrisation, or explicit knowledge of the goal distribution, isn't available? As the authors mention, there is some overlap with the work of Klink et al. (2019), by making the particular choice of f specified in Theorem 2. Can the authors comment on whether they expect other choices of f to be of interest? If I understand correctly, the remainder of the paper uses this choice of f. Theorem 1: If I understand the proof correctly, this additionally requires that the function f is chosen to be f(r) = exp(r/\eta). This should be added to the theorem statement. Theorem 2: The authors should perhaps mention that this choice of f is only possible with particular constraints on the MDP concerned (i.e. non-positive rewards is sufficient). Section 5 The lower plots of Figure 2 are hard to interpret as there is only one label on the y-axis. Table 1: What is meant by a p-test in the caption? Is this a typo for t-test? As discussed above, broader exploration of the performance of this approach/discussion of its applicability when low-dimensional context parametrisations are not available, or when the initial distribution over tasks is chosen poorly, would be interesting and strengthen the experiments section of the paper. Appendix Line 25: change \ref to \eqref. Minor: Reference formatting should be tidied up, e.g. capitalization of acronyms in article titles. For function definitions, \rightarrow should be used rather than \mapsto.


Review 3

Summary and Contributions: After reading the authors response, I've updated my score from (4) to (5). -------------------------------------------------- The paper proposes a curriculum learning algorithm for RL, SPDL. A fixed set of curriculum tasks is given, and the algorithm can sample tasks from the set at every step. The hope is that by smartly and adaptively selecting the tasks, we can speed up learning. The final goal is to maximize performance with respect to a fixed target distribution over tasks (which is known). The proposed algorithm alternates two types of steps: policy improving for a fixed task (or "context") distribution, and "task distribution adjustment" for a fixed policy. The former can be solved by means of any standard RL algorithm (as we can extend the MDP to account for the context). In order to solve the latter, the paper proposes a regularized objective (equation (3)) where we trade-off having high-reward given the current policy (first term) and not being very far from the target distribution of tasks (second term, KL). The trade-off is controlled by some hyper-parameter \alpha. Let w be the weights of the RL policy (the policy may depend on the context c), and let v be the weights parameterizing a distribution over contexts. The paper then tries to make some theoretical connections with RL-as-inference, where an optimality event 'O' is assigned a likelihood monotone on the cumulative discounted reward (see equation (4)). Motivated by maximizing this optimality probability with respect to the distribution over tasks, some equivalence results are presented in Theorem 1 and 2. Building on the fact that most RL algorithms to optimize weights w will provide an explicit value function, equation (5) provides the main element of the algorithm: the optimization problem from which we update the distribution over contexts (i.e., parameters v). In Section 5, the paper presents experiments on three continuous control settings. Two of them have dense rewards and a fairly "narrow" target task, whereas the third one provides sparse rewards but a wide range of target contexts. The experiments compare the proposed methods with other previously published methods, and with two basic baselines: sampling uniformly at random from the space of tasks ('random'), and sampling from the target distribution ('default'). In the dense-reward scenarios, the proposed method strongly outperforms the other methods (Figure 2, and Figure 3a, Table 1). In the sparse-reward scenario, results are more balanced, but SPDL still manages to beat the rest, especially when using PPO for w optimization.

Strengths: The algorithm seems simple to implement as, in order to optimize policy parameters w, we can plug in any standard RL algorithm that offers a value function estimate. The performance of the algorithm in the presented experiments is solid. In addition, inspecting the selected distribution over tasks could provide some interpretability on where the algorithm is focusing its learning at each training stage. The authors plan to open-source the code.

Weaknesses: Overall (unless I'm missing something), I find the theoretical results provided in Section 4 to be fairly disconnected from the final algorithmic design. Moreover, as the connection is not obvious, those results and comments make the paper more obscure and harder to parse. My understanding is that "Interpretation as Inference" and Theorem 1 are just applied to motivate the constraint in equation (5). I think this can be motivated with simpler arguments, like the variance of the importance-sampling estimate in the first term of (5) exploding if subsequent task distributions are very different. The paper makes several comments of future uses of the theoretical results to further tune some hyperparameters. Personally, I don't think this is a strong or material enough connection to place those results in the main text. Also, it's not clear to me how the algorithm would perform or work in the most natural setup where the set of tasks is discrete (and maybe even small), and the target task is a Dirac delta distribution over one of them. In this case, formally, even the KL term in (3) could be infinite. There's a line of work ([1] for example) where tasks are sampled proportionally to some learning proxies that measure how much we learn about the task of interest by solving/playing other tasks. I imagine the target scenarios for SPDL are (continuous control?) problems where defining / parameterizing a continuous set of *no-harder* tasks related to the final goal is easy (as the ones in the experimental section). For example, how would you go about solving some Atari game via SPDL? I think this is completely fine, but it should probably be explicitly stated upfront. What if the tasks are just a small set of (strategically selected) initial states from which we are allowed to reset the scenario --but we still only care at the end about one fixed s_o? Do we expect SPDL to work well? [1] Automated Curriculum Learning for Neural Networks - Alex Graves, Marc G. Bellemare, Jacob Menick, Remi Munos, Koray Kavukcuoglu.

Correctness: I did not check the proofs in the appendix. The experimental methodology seems reasonable.

Clarity: The paper is well written and easy to follow. I'm not sure the digression about variational inference helps the reader much.

Relation to Prior Work: The paper cites and compares to previous algorithms and methods.

Reproducibility: Yes

Additional Feedback: A couple of questions: - How do you estimate a good value / set epsilon in equation (5)? How sensitive is the algorithm to its value? - The first term in (5) could have a large variance -- this is also related to the value of epsilon. Have you observed anything like it in practice? What value of K do you use? Can you share results of the algorithm for several values of K? How does the training time increase with K?


Review 4

Summary and Contributions: The paper builds on prior work to present a method for curriculum learning that utilizes an EM-like optimization approach. The objectives that the algorithm is trying to optimize are 1) reward and 2) the negative KL of the current task distribution to the target task distribution. This is not new, but the authors present and claim better results by alternating optimization between the two in a block-coordinate ascent manner instead of optimizing them simultaneously.

Strengths: This is good work. The main strengths are that the paper does almost everything I expect it to to answer whether their method is more worthwhile than other methods. They present an interesting addition to curriculum learning in RL; they then carry out experiments comparing the proposed method to other approaches; and they do this on a diverse set of environments.

Weaknesses: The only thing that I felt lacking was an analysis of the actual curricula learned. I was looking forward to that and have a number of questions about it, e.g.: 1. Is it consistent in a task, i.e. will running this on a task consistently produce the same learning sequence? 2. Is independent of the underlying algorithm (PPO, SAC, ...)? 3. How sensitive is it? If you got hte curriculum and then changed some part of it and trained a new agent on this new sequence, how would it do? 3.

Correctness: Yes, they appear to support each other.

Clarity: Please do another pass through the paper. It was very easy to spot typos and other jarring mistakes that take the reader out of the flow. For example, there are two on L285 along: "the ball is said to be *catched* if it is in contact with the *endeffector* that is attached to the robot".

Relation to Prior Work: Yes. the main comparison is with Klinkle et al, and they demonstrate how their EM algorithm differs in theory, explanation, and empirical studies.

Reproducibility: Yes

Additional Feedback: UPDATE AFTER REBUTTAL: My score stands. I think it's a good submission and at least a solid accept. The authors addressed my concerns, but not to the extent that it would change my score.

[Author Response · NeurIPS 2020]

We would like to thank all reviewers for their feedback, hints towards typos as well as improvements in notation. We
will make sure to incorporate them in the final version of the paper. Reviewer comments are in **bold**.

**R1: Unclear how to choose the penalty proportion $\xi$, nor how it was selected. $N_\alpha$ not explained in the text.** $\xi$
has a monotone behavior, controlling the average progression speed to the target distribution. In all experiments, the
performance initially increased with an increase of $\xi$ until reaching a maximum. Hence a simple line search always
sufficed to find good values of $\xi$. We will move the discussion from the appendix to the main paper and ensure that $N_\alpha$
does not only appear in the algorithm box.

**R2: SPDL requires explicit knowledge of the desired distribution over contexts.** While this may seem a burden,
it actually makes assumptions of existing CRL algorithms explicit. Many CRL algorithms (including GoalGAN and
ALP-GMM) choose tasks for training based on a proxy for learning progress. In the absence of unlearnable tasks,
such an approach explores the whole context space, comparable to a uniform target distribution. This explains the
comparatively strong performance of SPDL compared to GoalGAN and ALP-GMM in Experiments 1 and 2, where only
a specific task is to be learned. ALP-GMM and GoalGAN cannot exploit this additional knowledge. **SPDL requires a**
**parametrization of the context space space (unlike e.g. GoalGAN).** Access to a parameterized context space is an
assumption shared by all CRL algorithms known to us (at least for continuous task spaces). In the experiments of the
GoalGAN paper, desired positions to be reached by the agent are used as parameterizations of the context space, as can
be verified in the code accompanying the GoalGAN paper. **Theorem 1 additionally requires that $f(r) = \exp(r/\eta)$.**
We indeed forgot the assumption $f(r) = \exp(r/\eta)$ in Theorem 1 placing it only in Theorem 2 and will correct the
mistake. Luckily, the assumption is standard in the RL-as-Inference literature so that its presence does not impair the
applicability of our algorithm. **Theorem 2: Choice of f is only possible with particular constraints on the MDP (i.e.**
**non-positive rewards is sufficient).** We think there is a misunderstanding. As correctly mentioned, for a probabilistic
interpretation of the rewards, these need to be assigned non-negative real numbers. This is, however, already enforced
by the monotonic transformation $f$ whose domain are the positive real numbers (introduced in line 127). $f$ represents
the (unnormalized) probability $p(\mathcal{O}|\tau, \mathbf{c})$. **What is meant by a p-test in the caption?** We will correct this typo to a
t-test. **Is SPDL applicable when a low-dimensional context parameterization is not available?** As mentioned, a
parameterization of the context space is unavoidable in the continuous setting. Regarding dimensionality, we believe
that the method as it currently is will likely scale to 5- to 7-dimensional context spaces. For higher-dimensional spaces,
we believe that more advanced representations of the context distribution than an anisotropic Gaussian are required.

**R3: theoretical results provided in Section 4 disconnected from algorithmic design.** We agree that the algorithm
can be motivated from simpler arguments. However, our theoretical interpretation connects it to the rich literature of
inference. We strongly believe that the theoretical grounding, although harder to process, allows for a more rigorous
understanding of why the algorithm works, and hence is just as important as good algorithmic performance. Further, the
theoretical motivation differs a lot from the commonly employed proxies based on Intrinsic Motivation which makes it
additionally interesting. **How would SPDL perform on a set of discrete tasks with a Dirac delta distribution over**
**one of them (e.g. creating a curriculum over finitely many initial states)?** SPDL can also be applied in discrete
settings, since the (currently Gaussian) context distribution of SPDL can easily be replaced with a discrete one. The
KL-Divergence employed in the RL-as-Inference framework, however, does not allow for a Dirac-Delta to be used.
SPDL as of now could only employ a smoothed version of such a Dirac-Delta or a uniform distribution if desired. We
ran a small experiment on an $8 \times 8$ Grid-World in which two keys need to be collected to open doors for reaching a goal
position. Only a reward of 10 is given when reaching the goal. We selected 8 different starting states and ran SPDL as
well as EXP3.S with "Absolute Learning Progress" (ALP), a standard formulation of intrinsic motivation in RL e.g.
used by ALP-GMM, over 20 seeds to generate curricula over the starting states. Using PPO, both SPDL and Exp3.S
generate curricula which prioritize states in a reverse order, i.e. moving from states close to the goal to the state furthest
away from it. Both algorithms improve upon uniform sampling over contexts, although EXP3.S performs slightly better
than SPDL. We are convinced that there are many possibilities for improving the algorithmic implementation of SPDL
in discrete scenarios, as the current implementation focuses on continuous context spaces (e.g. for robotics) where a
big challenge are intractable integrals. In a discrete setting, expectations can often be easily evaluated, allowing e.g.
for more advanced sampling techniques from the context distribution. **How would you solve some Atari game via**
**SPDL?** For Atari games, a context space encoding environment versions of different difficulty could be defined. For
Space-Invaders an interesting parameterization could be movement speed and shooting frequency of enemies.

**R4: Is [the curriculum] consistent in a task/independent of the underlying algorithm? How sensitive is it?** We
re-investigated the curricula generated by SPDL in the experiments. The evolution of the distributions looks consistent
across algorithms. Depending on the task, they however exhibit a significant amount of variance, as e.g. in the point
mass experiment: While the friction parameter is continuously annealed (although with varying pace), there are curricula
that prioritize moving the gate to the target position before shrinking it to target size or, vice versa, first shrink the gate
and then move it. Further, there exist all interpolations between these two extremes. Investigations of sensitivity were
unfortunately out of scope given the short time of the rebuttal period, but are certainly interesting points for future work.

[Meta-Review · NeurIPS 2020]

This paper presents a method for curriculum generation in reinforcement learning, by shaping the sampling distribution in a dynamic way to improve performance on a target task distribution. There is clear intuition and exposition of the method, and a good evaluation on a variety of environments and RL algorithms showing positive results. I encourage the authors to incorporate the feedback of the reviewers in their final draft.